# Glycaemic and Appetite Suppression Effect of a Vegetable-Enriched Bread

**DOI:** 10.3390/nu13124277

**Published:** 2021-11-27

**Authors:** Isaac Amoah, Carolyn Cairncross, Fabrice Merien, Elaine Rush

**Affiliations:** 1Faculty of Health and Environmental Sciences, Auckland University of Technology, Auckland 1142, New Zealand; carolyn.cairncross@aut.ac.nz; 2Centre of Research Excellence, Riddet Institute, Palmerston North 4474, New Zealand; 3Department of Biochemistry and Biotechnology, Kwame Nkrumah University of Science and Technology, Kumasi 0023351, Ghana; 4AUT Roche Diagnostics Laboratory, Auckland University of Technology, Auckland 1010, New Zealand; fabrice.merien@aut.ac.nz

**Keywords:** vegetable-enriched bread, white bread, wheatmeal bread, serum glucose, insulin response, appetite suppression

## Abstract

Bread, a frequently consumed food, is an ideal vehicle for addition of ingredients that increase nutrient density and add health benefits. This experimental cross-over study sought to test the effect of a vegetable-enriched bread (VB) in comparison to commercial white bread (WB) and wheatmeal bread (WMB) on serum glucose, insulin response and subjective appetite suppression. On three separate occasions, 10 participants (23 ± 7 years) visited the laboratory and consumed after an overnight fast, in random order, a 75 g serve of WB, WMB or VB. Venous blood samples drawn twice before (0 min) and at 15, 30, 45, 60, 90 and 120 min after consumption of the bread were analysed for glucose and insulin. Participants rated their subjective feelings of hunger, fullness, satisfaction and desire to eat on a 150 mm Likert scale. The mean glucose iAUC over 120 min was not different among the breads. The mean insulin iAUC for the VB was significantly lower than the WB and WMB; difference VB and WB 12,415 pmol/L*minutes (95% CI 1918, 22,912 pmol/L*minutes, *p* = 0.025) and difference VB and WMB 13,800 pmol/L*minutes (95% CI 1623, 25,976 pmol/L*minutes *p* = 0.031). The VB was associated with a higher fullness feeling in the participants over the 120-min period. The consumption of VB was associated with less insulin release and higher satiety over 120 min which may be related to the higher fibre content and texture of VB. The role of vegetable and fruit fibres such as pectin in bread and insulin response should also be further explored.

## 1. Introduction

Globally, the quality and quantity of the food supply, is a major attributable factor for the development of diet-related non-communicable diseases including type 2 diabetes mellitus [1]. In 2009, bread and bread products contributed the highest proportions of energy (11%), protein (11%), carbohydrate (17%), and dietary fibre (17%) to the New Zealand diet [2]. The raw material used in bread formulation may impact on its glycaemic response [3] and subjective appetite sensations [4]. White breads formulated from refined flours, for example, are usually poor sources of dietary fibre, elicit a higher glycaemic response [5], and have a lower ability to suppress appetite sensation [4]. Wholegrain and higher fibre breads, with a lower glycaemic index, are sought by health–conscious consumers [6]. In New Zealand, wheatmeal bread is a low cost and frequently sold bread that contains milled whole wheat [7]. The addition of novel ingredients to increase dietary fibre content in bread is known to favourably impact on glycaemic response and satiation [8].

Novel ingredients that improve glycaemic response and appetite suppression include brown sorghum [9], hazelnuts [10], wholemeal rye flour [11], salba seeds [12,13] and chickpea flour [14]. What is not known is whether drum-dried vegetable flours [15] with low moisture content and long shelf-life, and nutrient-enriching and phytochemical properties when used to enrich bread, will demonstrate effects that could support a health claim. We have already shown [16] that bread with ingredients selected for wholesomeness and nutritional quality—drum-dried pumpkin and sweet corn flours, sprouted wheat flour, wholemeal wheat flour and flaxseed—is acceptable to consumers and easier to swallow than commercial white and wheatmeal breads.

Pumpkin flour is a good dietary source of carotenoid compounds, particularly beta carotene, and also the hydrophilic fibre pectin [17,18]. Sweet corn flour is a rich source of the dietary fibres, cellulose and β-glucan, which enhance its ability to promote satiety [19]. Flaxseed and wholemeal flour have shown suppression of postprandial appetite [20] and improved glycaemic response [21]. Another novel ingredient was sprouted wheat. Germination of grains increases antioxidant activity [22]. Recently, sprouted wheat has been found to function as an enzyme improver in bread [23] and to lower the glycaemic index [24].

A strong association between the consumption of vegetables and lower incidence of diet-related non communicable diseases including hypertension and type 2 diabetes mellitus has been established in two independent meta analyses of prospective cohort studies [25,26]. However, vegetables including pumpkin and sweetcorn are seasonal and highly perishable due to their high moisture content and water activity [27]. This warrants that they are transformed into flours that have a long shelf life such as drum dried flours. A review of published original articles on the two prominent electronic databases PubMed and Web of Science showed that no study has incorporated drum-dried pumpkin and sweetcorn powders into bread and its glycaemic and satiety responses investigated. The validation of the glycaemic and satiety effect associated with the consumption bread formulated with drum-dried pumpkin and sweet corn flours could promote a pathway for increased and sustainable consumption of vegetables throughout the year and reduce vegetable loss and waste. The study therefore sought to test the effect of this vegetable-enriched bread, in comparison to commercial white and wheatmeal breads, on serum glucose, insulin response and subjective appetite suppression effect over a 120-min period.

## 2. Materials and Methods

### 2.1. Participants

The experimental study was approved by the Auckland University of Technology Ethics Committee (AUTEC, approval number 18/147). The study was advertised at the AUT South Campus. Ten apparently healthy participants who were relatively sedentary and regular bread eaters (at least three times per week) provided written, informed consent. Exclusion criteria included regular and high intensity activity, following a low carbohydrate diet and gluten intolerance. The establishment of health claims related to the reduction in postprandial glycaemic response and that the test bread has a statistically significant decrease (minimum 20%) in incremental area under the blood glucose response curve (iAUC) in comparison to the reference white bread [28]. Mean coefficient of variations (CVs) of the iAUC for glucose are reported in the range of 20–30% [29]. Therefore, in this study for a predicted minimum 20% decrease in iAUC with a CV of 20%, it was calculated (G*power 3.1, Heinrich Heine University, Germany) that 10 participants would be required to show a 20% difference in glucose iAUC between treatments for a power of 80% with an alpha of 0.05.

### 2.2. Study Design and Methods

Participants were asked to follow their normal evening meal and then to fast overnight for at least eight hours before visiting the laboratory at 9 a.m. the next day. Consump-tion of water was allowed. The participants were additionally asked to restrict the intake of alcohol and excessive physical activity the day before each test.

In this experimental, cross-over study, participants in a predetermined random order and on three separate occasions separated by at least a three-day interval consumed 75 g of white bread (WB), wheatmeal bread (WMB), or vegetable-enriched bread (VB). Six different sequences of presentation of the three breads were possible. Two blocks of the six sequences (i.e.,12 sequences) were randomly sorted using the Excel ™ random number generator. This predetermined order was applied in order of presentation of the participant at the laboratory for their first visit. The participant was not told which order they would receive. The breads were frozen at −16 ℃ but defrosted one hour prior to consumption. The breads were served to participants, with 250 mL of tap water. The participants were not blinded as they could see and taste what they were eating. The white bread could clearly be differentiated by colour. However, the breads were not labelled to reduce the risk of biases. A venous catheter (BD Venflon, Becton Dickinson, Helsingborg, Sweden) was inserted into the antecubital vein. Two fasting blood samples were taken five minutes apart and then the participants were asked to consume the 75 g of bread within 12 min. The subjects were seated comfortably and listened to music of their own choice throughout the procedure.

### 2.3. Measurements

Venous blood samples were drawn for the analysis of serum glucose and insulin before the meal (0 min) and at 15, 30, 45, 60, 90 and 120 min after consumption of the bread.

The drawn blood sample stood for 20–40 min and then was centrifuged for 10 min at 3500 rpm (centrifuge Z 150 A, Woodbridge, NJ). Serum was then aliquoted into Eppendorf tubes and immediately frozen at −20 ℃. The frozen serum was transported in ice to the AUT Roche Diagnostics Laboratory. Glucose and serum concentrations were determined by specific diagnostic assays on an automated clinical chemistry analyser (Cobas Modular P800/E170, Roche Diagnostics New Zealand Ltd., Auckland, New Zealand). The test principle for the glucose measurement was by enzymatic colorimetric test with limits of measurement being 0.11–41.6 mmol/L. For the insulin measurement, the test principle employed was electrochemiluminescence with limits of measurements 1.39 to 6945 pmol/L. For all assays, the limit of measurement or functional sensitivity represents the lowest measurable analyte level that can be distinguished from zero. Method comparisons, limitations and specific performance data can be found at www.e-labdoc.roche.com (accessed 8 November 2021).

### 2.4. Appetite Assessment

The participants were asked, at each timepoint when the blood was drawn, to rate their subjective feelings of how the bread impacted on their appetite by marking a line on a 150 mm visual analogue scale (VAS) before and after eating the bread. The visual analogue scale had questions relating to appetite sensations and their corresponding anchor words: Are you hungry? “Not hungry” and “Extremely hungry”, Could you eat? “Can eat more” and “Too full, cannot eat anymore”, Are you satisfied? “Less satisfied” and “Extremely satisfied”, and Do you want to eat? “Less desire to eat” and “Extreme desire to eat”. The distance on the scale from the left-hand anchor point was measured with an electronic digital caliper (Warrior^®^, Cornwall, UK) to a precision of 0.01 mm.

### 2.5. Preparation and Nutritional Characterisation of Breads

The vegetable-enriched bread was prepared according to the procedure used in our previous work [16]. The vegetable enriched bread was formulated from the ingredients: strong white wheat flour, wholemeal wheat flour, whole flaxseed, sprouted wheat flour, pumpkin flour, sweet corn flour, fresh yeast and salt. The ingredients are listed in a descending order by weight. The indirect method of bread-making was utilised for the bread production and the procedure employed is briefly described below. A pre-ferment that consisted of 325 g wholemeal wheat flour, 150 g water, and 0.5 g instant yeast was developed for 2 min to obtain a tight dough consistency. Fermentation of the mixture was allowed to proceed at 20 ℃ with the mixture covered for 12 h. Whole flaxseed of weight 150 g was immersed in boiling water and then left at room temperature for 2 h before placement in a chiller for 10 h. The final dough was formulated by mixing 450 g strong wheat flour, sprouted wheat flour (15 g), pumpkin powder (75 g), sweet corn powder (20 g), salt (15 g), instant yeast (5 g) and water (600 g) for 8 min. The dough development was allowed to proceed for 6 min. The soaked whole flaxseed was the last ingredient added and the dough was thoroughly mixed. The dough underwent bulk fermentation for one and half hours. The fermented dough was cut and placed in baking pans and allowed to prove for an hour. The dough was baked in a steam oven at 215 ℃ for 35 min. The loaves were allowed to cool and packaged in transparent packs.

Selected nutrients in the vegetable-enriched bread (Table 1) were determined at Asurequality, an Internationally Accredited New Zealand laboratory. Analyses were moisture content (AOAC 925.10), β-carotene (EN 12823–2:2000, COST91, 1986), dietary fibre (insoluble) (AOAC 991.43 Ankom), dietary fibre (soluble) (AOAC 991.43 Ankom), dietary fibre (total) (AOAC 991.43), sodium (AsureQuality Method, ICP-OES) and potassium (AsureQuality Method, ICP-OES). The HSRCS (% foods method) for the VB, WB and WMB was calculated in compliance with FSANZ regulations [30].

### 2.6. Statistical Analyses

Data was checked for normal distribution using the Shapiro–Wilk test (where *p* > 0.05 implied normality for the data) due to the small sample size. The total incremental area under the curve (iAUC) was calculated for both glucose and insulin responses following WB, WMB and VB consumption. The calculation followed the ISO standard for glycaemic index and applied the trapezoid rule (The International Organization for Standardization, 2010). The area of the “curve” above the fasting baseline was assumed to be a trapezoid and calculated as the sum of the areas of a triangle and a rectangle with the units for glucose of mmol/litre* minutes. Differences among breads were determined by the general linear model, repeated measures and analysis of covariance for the total iAUC of the glucose and insulin responses. In the case of appetite responses including “Are you hungry?”, “Could you eat?”, “Are you satisfied?” and “Do you want to eat?” differences among breads were determined by the general linear model, repeated measures and analysis of covariance of the change from baseline. Unless otherwise stated, all data were analysed using International Business Machines Corporation^®^ SPSS^®^ Statistics Version 25.

## 3. Results

### 3.1. Nutrient Analysis

Bread (VB) enrichment increased the nutrient density resulting in higher dietary fibre, potassium and β-carotene compared to the WB and WMB (Table 2). White bread (WB) had a lower fibre content compared to the VB and the WMB. The WB and WMB breads were also higher in sodium content compared to the VB bread.

### 3.2. Participants

All of the ten apparently healthy participants (9 males, 1 female) mostly of Pacific ethnicity (*n* = 8) with an average age of 23 ± 7 years and BMI of 32.1 ± 4.5 kg/m^2^ completed all tests (Table 3).

### 3.3. Glucose and Insulin Response

Generally, after consumption of each bread, the glucose and insulin concentrations were at their highest peak between 15 and 60 min and below baseline at 120 min, which was associated with an increase in hunger. Consumption of the VB elicited a glucose response that was not different from the WB and WMB (Table 4 and Figure 1). The mean difference of the iAUC for the VB compared with the WB was 19.1 mmol*min/L (95% CI −14.8, 53.0 mmol*min/L, *p* = 0.235) and WMB was 10 mmol*min/L (95% CI −18.9, 39.4 mmol*min/L, *p* = 0.449).

The consumption of the WMB and WB bread elicited significantly higher insulin responses than VB (Table 5 and Figure 1). The mean insulin iAUC for the VB was almost 38% less than that of the other two breads. The mean difference of the insulin iAUC for the VB compared with the WB was 12,415 pmol*min/L (95% CI 1918, 22,912 pmol*min/L, *p* = 0.025) and WMB was 13,800 pmol*min /L (95% CI 1623, 25,976 pmol*min/L, *p* = 0.031).

Throughout the 120 min the rating of hunger was lowest for VB, while participants reported only a small initial decrease in hunger up to 60 min for the WMB, from which point hunger increased (Figure 2). WB had the least hunger suppressing effect. The mean difference of the change in baseline of the VB compared with the WB and WMB was WB 80.29 mm (95% CI −65.18, 225.75 mm) and WMB 52.91 mm (95% CI −111.60, 217.41 mm). There were, however, no significant differences among the hunger suppression effects of the VB, WMB and WB.

The VB had a higher appetite satisfaction compared to the WB and WMB (Figure 2). A significant difference existed in the ability of the VB to provide satisfaction compared to the WB and WMB. The VB was rated by the participants as filling and thus they were not wanting to eat more, particularly at the 15th and 30th minute measurement points.

The vegetable-enriched bread impaired the participants’ ability to eat more than the other two breads. The mean difference of the change in baseline of the vegetable-enriched bread compared with the WB and WMB was WB −27.49 (95% CI −94.42, 39.45) and WMB −79.64 (95% CI −213.22, 53.94).

## 4. Discussion

This study has shown that bread enriched with drum-dried pumpkin and sweet corn powders significantly attenuated insulin release by 38% compared with the white and wheatmeal breads but with no apparent difference in glucose release. We were able to only find one other similar observation from Coe and Ryan [32] who reported that a powder of the baobab fruit added to white bread was not associated with a reduction in glucose release and satiety over a 180-min period but, similar to this investigation, insulin iAUC was attenuated by about 20%. The authors attributed this observation to the polyphenol compounds in baobab that may reduce the rate and degree of starch digestibility. It is of interest that baobab fruit, like pumpkin, is a source of source of the dietary fibre pectin.

Our study and the one of Coe and Ryan [32] demonstrate that the addition of vegetables and fruit are associated with insulin but not glucose release attenuation. It is already known that bread made with refined flour only, i.e., low fibre, elicits a higher (~50%) insulin iAUC concentration compared with wholemeal and grain breads but glucose iAUC for the white bread is 40% higher for refined flour bread [33]. In another study of grain products, glucose iAUC over a 180-min period to consumption of white wheat bread, whole-kernel rye bread, oat β-glucan-enriched rye bread and wholemeal pasta were not different [34]. Gastric emptying time was not different. However, compared to the white bread 30 min postprandial insulin release was reduced by 40% and 20% for the whole-kernel rye and β-glucan-enriched rye breads respectively. The whole-kernel rye and β-glucan-enriched rye breads had a fibre content of 12.8 and 17.1% respectively which does not explain the difference in insulin response. The authors posited that structural properties and matrix of the grain products affected the rate of insulin response [34].

As postulated by others, the reduction in the insulin response to the vegetable-enriched bread following consumption could be attributed to three factors. The first and second factors may be related to the higher fibre and viscous matrix of the VB, which may reduce the rate of bread bolus digestion [33], and the particle size and shape of ingredients [34] mediating the release of the incretin hormones glucose-dependent insulinotropic polypeptide (GIP) and glucagon-like peptide (GLP) [35] which in turn reduce the secretion of insulin. Although in this present study, incretin hormones glucose-dependent insulinotropic polypeptide (GIP) and glucagon-like peptide (GLP) were not measured, it could however be considered in future investigations. Pectin fibres in pumpkin [18] and gums and mucilage from flaxseed [20,36] synergistically enhance the viscous fibre composition of the VB. The consumption of flaxseed gum enriched-chapatti and flaxseed mucilage-enriched meals resulted in significant attenuation of insulin release [20,36].

The last factor that has been implicated in the modulation of glycaemic control includes the presence of polyphenols which would be derived from the pumpkin, corn and sprouted wheat, inhibiting the action of carbohydrate-digesting enzymes and reducing the rate of digestion [37].

A plausible explanation for the postprandial appetite satisfaction sensation elicited by the VB could be its high fibre content due to the pumpkin [18], sweet corn and flaxseed enrichment [20,36]. Fibre impacts on the matrix and texture of the enriched bread [38] and increases chewability, consequently inducing appetite suppression sensations [39]. In a previous study by the same authors where the “Swallowing and liking of vegetable-enriched bread compared with commercial breads as evaluated by older adults” was investigated, the authors reported that, bread enriched with vegetables was easier to chew and swallow than the commercial breads [16]. Additionally, due to the high fibre composition of VB, its consumption may increase gastric emptying time and increase the bulk in the gastrointestinal tract [8]. This eventually results in the secretion of gut hormones including PYY, cholecystokinin (CCK), GLP-1, and pancreatic polypeptide (PP) [40,41]. The released hormones pass through the blood-brain barrier, bind to receptors in the satiety centres of the hypothalamus in the brain and trigger a satiety signalling cascade that makes the participants feel full. Associations between the release of appetite-suppressing hormones and insulin release regulation and glycaemic control has also been reported. Ibrügger et al. [42] investigated how the consumption of flaxseed dietary fibre supplements acutely impact on appetite and suppress food intake. The study was in two-fold and employed a single-blinded randomised crossover design where 24 and 20 participants respectively took a 300 mL drink (control) and a 300 mL with added flax fibre extract (2.5 g of soluble fibres) (first study) and control drink with flax fibre tablets (2.5 g of soluble fibres) (second study). The authors reported that the consumption of the flaxseed drink resulted in an increased fullness sensation compared to the control, consequently leading to a significant decrease in subsequent energy intake.

The participants who participated in the appetite study had a body mass index which is categorized as obese (Table 2). In children who are obese, it has been reported that there is an increased desire to eat compared with those categorised as non-obese [43]. The large body size of the participants could be related to the inability to differentiate between the appetite attributes of the relatively small serving (75 g) of bread and more investigation with larger doses of bread should be investigated to determine effects on appetite. However the vegetable bread did elicit a higher sensation of fullness.

### Strengths and Limitations

The principles of the protocol for glycaemic index testing [44] and calculation of the incremental areas under the curve were followed, with the number of participants (10) meeting the requirements for sample size. The food portion size was not based on a standard amount of carbohydrate (50 g) but on a realistic serving size of each bread (75 g). Measurement error was minimised with analysis of glucose and insulin on all stored samples undertaken on one day in one run with an accredited medical laboratory system.

One limitation of the work is that the bread types used differed from each other in terms of their ingredient composition. In traditional food product development work, the bread would have been formulated with variation in one ingredient at a time to determine its effect on the glycaemic and appetite response when consumed. However, to make the study practical in its approach and as the vegetable bread had already shown that it was more liked, the research question was limited to the comparison of the three breads. The ethnicity, body size, age, and sex of the participants was also limited.

## 5. Conclusions

This investigation is a proof-of-principle comparison of glucose and insulin responses to two popular commercial breads, white and wheatmeal breads, and a novel vegetable-enriched bread. Consumption of the vegetable-enriched bread was associated with a postprandial reduction in insulin released and an increased feeling of fullness compared with commercial breads. Future studies could better define the participants’ glycaemic status and could compare the response of those previously diagnosed with and without type 2 diabetes mellitus. The addition of vegetables, fruit and seeds to bread and the effect of fibres such as pectin on insulin response should also be further explored.

## Figures and Tables

**Figure 1 nutrients-13-04277-f001:**
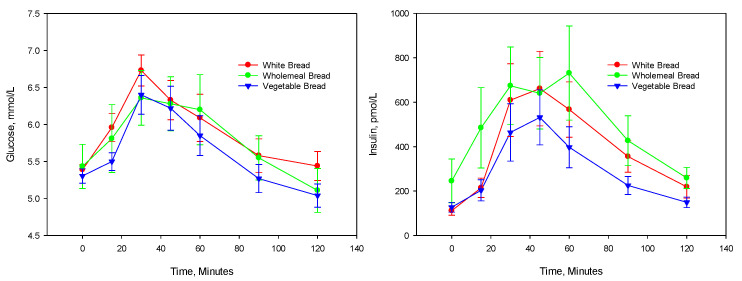
Mean (SEM) responses (*n* = 10) of glucose and insulin following consumption of 3 breads.

**Figure 2 nutrients-13-04277-f002:**
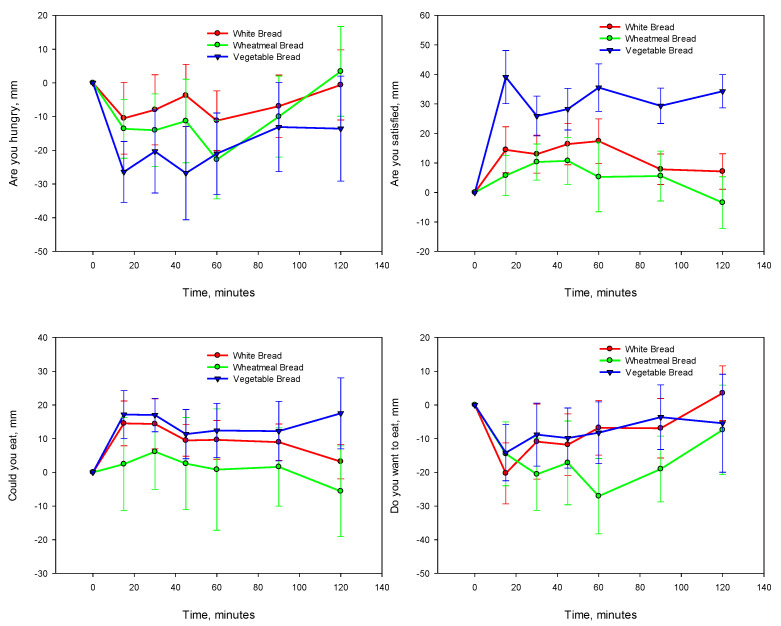
Changes in satiety responses (*n* = 10) from baseline following the consumption of breads. Mean SEM error bars.

**Table 1 nutrients-13-04277-t001:** Ingredients used in white bread, wheatmeal bread and vegetable-enriched bread formulation.

White Bread *	Wheatmeal Bread *	Vegetable Bread
Wheat flour	Wheat flour	Water
Water	Water	Wheat flour
Baker’s yeast	Wheatmeal flour	Wheatmeal flour
Iodised salt	Baker’s Yeast	Flaxseed
Canola oil	Vinegar	Pumpkin powder
Acidity regulator (263)	Iodised salt	Sweet corn powder
Soy flour	Wheat gluten	Sprouted wheat flour
Emulsifiers (481, 472e)	Acidity regulator (263)	Salt
Vitamin (Folic Acid)	Roasted barley malt flour	Baker’s yeast
	Canola oil	
	Soy flour	
	Emulsifiers (481, 472e)	
	Vitamin (Folic Acid)	

* as listed on the food label of the commercial bread. Ingredients are listed in descending order by weight.

**Table 2 nutrients-13-04277-t002:** Nutrient composition of the test breads.

Component	White Bread *	Wheatmeal Bread *	Vegetable Bread †
Moisture (g/100 g)	36.6	38.2	39.1
Protein (g/100 g)	8.5	8.8	7.5 *
Dietary fibre (g/100 g)	2.7	4.6	7.2
Insoluble fibre (g/100 g)	nd	nd	5.5
Soluble fibre (g/100 g)	nd	nd	1.7
Fat (g/100 g)	1.6	1.7	4.8 *
Carbohydrate (g/100 g)	46.7	43.1	33.9 *
Sodium (mg/100 g)	392	398	380
Potassium (mg/100 g)	nd	nd	300
Energy (kJ/100 g)	1020	982	932 *
β-Carotene (µg/100 g)	nd	nd	236.8

* from nutrition information panel of food label of commercial bread † analysis by AsureQuality, an internationally accredited New Zealand Laboratory, nd = not determined.

**Table 3 nutrients-13-04277-t003:** Participant characteristics at baseline (*n* = 10).

Measure	Unit	Mean	Standard Deviation	Range	Interquartile Range
Age (years)	years	23.1	7.0	23	4
Weight	kg	106.7	18.0	60	12.8
Height	cm	181.9	11.0	38	15.8
BMI	kg/m^2^	32.1	4.5	15	7.21
Glucose	mmol/L	5.3	0.4	1.75	0.33
Insulin	pmol/L	131	69.6	215	133.1
HOMAR-IR	mmol/L*pmol/L	31	16.7	52.4	32.7
HOMAR-%B	p/mol/mmol	1498	957	4794.6	1139.7

BMI = body mass index, HOMA1-IR = homeostatic model assessment-insulin resistance, HOMA1-%B. = homeostatic model assessment-insulin resistance-beta cell function, calculated using the formula from Wallace et al. [31].

**Table 4 nutrients-13-04277-t004:** Participants’ area under curve, serum glucose after 120 min for test breads.

Bread	Mean/mmol*min/L	Standard Error	95% Confidence Interval	
			Lower Bound	Upper Bound
White bread	75.9	13.5	45.3	106.4
Wheatmeal bread	67.0	9.9	44.7	89.3
Vegetable bread	56.8	9.6	35.0	78.5

**Table 5 nutrients-13-04277-t005:** Participants’ serum area under curve, insulin after 120 min for test breads.

Bread	Mean/mmol*min/L	Standard Error	95% Confidence Interval	
			Lower Bound	Upper Bound
White bread	32,892	8058	14,663	51,121
Wheatmeal bread	34,276	8594	14,835	53,718
Vegetable bread	20,476	4425	10,466	30,488

## Data Availability

Owing to privacy commitments to participants in the approval by the ethics committee we cannot make the data publicly available, but please contact the author if further information is required.

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
