# Peer review of "Glycaemic and Appetite Suppression Effect of a Vegetable-Enriched Bread"

_nutrients, 2021, doi:10.3390/nu13124277_

Round 1
Reviewer 1 Report
The manuscript presented by Amoah et al. aimed to compare the effect of a vegetable-enriched bread (VB, commercial white bread (WB), and wheatmeal bread (WMB) on serum glucose, insulin response, and subjective appetite suppression.
Overall the manuscript is well organized and properly written, but in my opinion, the manuscript adds nothing new to science. Many studies proved the effect of vegetables (as a good source of fiber and bioactive substances) on insulin response, satiety, and glucose control. As the authors of the manuscript note:" The addition of novel ingredients to increase dietary fiber content in bread is known to favorably impact on glycaemic response and satiation [8]."
Below are my comments:
INTRODUCTION
The aim of the paper is clearly stated but the introduction does not provide enough information for readers to highlight the background of this study. Please provide more information to emphasize the significance of this study.
METHODS
The methods are overall appropriate for the aim of the study and sufficiently described. However, a small number of respondents draw attention. Were the sample size and power of the study estimated? „The participants were not blinded as they could see what they were eating” – the VB, WB and WMB looked the same?
RESULTS
The results are presented clearly in tables and figures. Table 3 – it is not clear what „t=0” means.
CONCLUSION
I suggest the authors should more highlight the novelty of the present study and distinction from previous studies.
In addition, some typos Leeds correction, e.g.:
- Page 1, line 44 – double space
- Page 3, line 101 -4
- Table 3, line 186 – unnecessary dot
Author Response
RESPONSE TO REVIEWERS COMMENT ON MANUSCRIPT WITH ID 1477716
Please find below our response to the reviewers comments on our manuscript entitled “Glycaemic and appetite suppression potential of a vegetable-enriched bread”.
Reviewer 1
Comments and Suggestions for Authors
The manuscript presented by Amoah et al. aimed to compare the effect of a vegetable-enriched bread (VB, commercial white bread (WB), and wheatmeal bread (WMB) on serum glucose, insulin response, and subjective appetite suppression.
Overall the manuscript is well organized and properly written, but in my opinion, the manuscript adds nothing new to science. Many studies proved the effect of vegetables (as a good source of fiber and bioactive substances) on insulin response, satiety, and glucose control. As the authors of the manuscript note:" The addition of novel ingredients to increase dietary fiber content in bread is known to favorably impact on glycaemic response and satiation [8]."
Response to comments:
Thank you for the comment. The authors agree with the reviewer that several functional ingredients have been used previously in bread formulation to improve its bioactive properties. However, to the best of our knowledge and checking from databases including PubMed and Web of Science, no work has utilised drum-dried flours from pumpkin and sweet corn in functional bread reformulation and its glycaemic and satiety effect investigated. This work comes to fill that gap.
Below are my comments:
INTRODUCTION
The aim of the paper is clearly stated but the introduction does not provide enough information for readers to highlight the background of this study. Please provide more information to emphasize the significance of this study.
Response to reviewer comments
Again the reviewers are grateful to the reviewer for suggesting we project the significance of the work. The authors have thus revised the introduction to capture this
A strong association between the consumption of vegetables and lower incidence of diet-related non communicable diseases including hypertension and type 2 diabetes mellitus has been established in two independent meta analyses of prospective cohort studies (Wang, Fang, Gao, Zhang, & Xie, 2016; Zurbau et al., 2020). However, vegetables including pumpkin and sweetcorn are seasonal and highly perishable due to their high moisture content and water activity (Maltini et al., 2003). This warrants that they are transformed into flours that have a long shelf life such as drum dried flours. A review of published original articles on the two prominent electronic databases PubMed and Web of Science showed that no study has incorporated drum-dried pumpkin and sweetcorn powders into bread and its glycaemic and satiety responses investigated. The validation of the glycaemic and satiety effect associated with the consumption bread formulated with drum-dried pumpkin and sweet corn flours could promote a pathway for increased and sustainable consumption of vegetables throughout the year and reduce vegetable loss and waste.
METHODS
The methods are overall appropriate for the aim of the study and sufficiently described. However, a small number of respondents draw attention. Were the sample size and power of the study estimated?
Response to reviewers comment
The sample size and power calculation is reported in Lines 72 to 80 under the section title “2.1. Participants”. We have edited this section for clarity
The establishment of health claims related to the reduction in postprandial glycaemic response and that the test bread has a statistically significant decrease (minimum 20%) in incremental area under the blood glucose response curve (iAUC) in comparison to the reference white bread [25]. Mean coefficient of variations (CVs) of the iAUC for glucose are reported in the range of 20-30% [26]. Therefore, in this study for a predicted minimum 20% decrease in iAUC with a CV of 20%, it was calculated (G*power 3.1, Heinrich Heine University, Germany) that 10 participants would be required to show a 20% difference in glucose iAUC between treatments for a power of 80% with an alpha of 0.05.
“The participants were not blinded as they could see what they were eating” – the VB, WB and WMB looked the same?
Response to reviewers comment
The reviewer is correct. The breads were physically different from each other. The white bread was different from the vegetable-enriched bread which had an orange crumb due to its carotenoid content whereas the wheatmeal bread crumb was brownish due to the bran enrichment. The manuscript has been amended to clearly state
The participants were not blinded as they could see and taste what they were eating. The white bread could clearly be differentiated by colour.
RESULTS
The results are presented clearly in tables and figures. Table 3 – it is not clear what „t=0” means.
Response to reviewer comments
In Table 3, t=0 meant baseline. However, because the Table title contains “baseline”, the authors have deleted the t=0. Consequently, the new Table title is “Participant characteristics at baseline (n=10).
CONCLUSION
I suggest the authors should more highlight the novelty of the present study and distinction from previous studies.
Response to reviewer comments
In the introduction and discussion we have now emphasized the novelty of the present study and emphasized the vegetable content of the bread. In particular:
Pectin, a component of dietary fibre of vegetables and fruits is present in the bread in the present study and also a bread containing Baobab where the observation of attenuation of insulin release but not glucose was undertaken.
We further suggest in the discussion that structural properties of the bread matrix may effect the insulin but not the glycemic response
This article is to be considered for the section: carbohydrates and we believe the distinction between the fibre of wheat as in the commercial breads and fibre in vegetables is appropriate for the siting of this paper in the discussion concerning carbohydrates in food.
In addition, some typos needs correction, e.g.:
- Page 1, line 44 – double space
Response to reviewer comment
This has been addressed
- Page 3, line 101 -4
Response to reviewer comment
Please, could the reviewer provide some form of clarification on what he or she means?
- Table 3, line 186 – unnecessary dot
Response to reviewer comments
The authors would like to explain that what appears as a dot is not a dot. The word there is fibre but was broken into two by the template properties and consequently appears as “fi-bre”.
The authors have also attached the response to the reviewers comment as a PDF document. "Please see the attachment."

Reviewer 2 Report
According to my knowledge, it is a novel paper in its field opening new horizons for further evidence. In addition, the object as well as the results are appropriately discussed in the context of previous literature explaining the importance of the manuscript in its field. Authors succeed to present their data in a clear way adding information to the existing literature.
Therefore, I have no corrections or further work to propose for the improvement of the manuscript .
Author Response
RESPONSE TO REVIEWERS COMMENT ON MANUSCRIPT WITH ID 1477716
Please find below our response to the reviewers comments on our manuscript entitled “Glycaemic and appetite suppression potential of a vegetable-enriched bread”.
Reviewer 2
Comments and Suggestions for Authors
According to my knowledge, it is a novel paper in its field opening new horizons for further evidence. In addition, the object as well as the results are appropriately discussed in the context of previous literature explaining the importance of the manuscript in its field. Authors succeed to present their data in a clear way adding information to the existing literature.
Therefore, I have no corrections or further work to propose for the improvement of the manuscript.
Response to reviewer comments:
The authors are grateful to the reviewer for the positive comments.
Please, the authors have additionally attached the response to the reviewer comments for your perusal. "Please see the attachment."

Round 2
Reviewer 1 Report
Thank you for the revisions to the manuscript.
Please note a minor editorial error in the references notation (Page 2, line 61).